# Bionic 3D printed corals

Daniel Wangpraseurt [1,2,3 ✉], Shangting You[4], Farooq Azam[2], Gianni Jacucci [1], Olga Gaidarenko [2], Mark Hildebrand[2,8], Michael Kühl [3,5], Alison G. Smith [6], Matthew P. Davey [6], Alyssa Smith[1], Dimitri D. Deheyn [2], Shaochen Chen [4,7 ✉] & Silvia Vignolini [1,7 ✉]

Corals have evolved as optimized photon augmentation systems, leading to space-efficient microalgal growth and outstanding photosynthetic quantum efficiencies. Light attenuation due to algal self-shading is a key limiting factor for the upscaling of microalgal cultivation. Coral-inspired light management systems could overcome this limitation and facilitate scalable bioenergy and bioproduct generation. Here, we develop 3D printed bionic corals capable of growing microalgae with high spatial cell densities of up to $10^9$ cells mL$^{-1}$. The hybrid photosynthetic biomaterials are produced with a 3D bioprinting platform which mimics morphological features of living coral tissue and the underlying skeleton with micron resolution, including their optical and mechanical properties. The programmable synthetic microenvironment thus allows for replicating both structural and functional traits of the coral-algal symbiosis. Our work defines a class of bionic materials that is capable of interacting with living organisms and can be exploited for applied coral reef research and photobioreactor design.

[1] Bioinspired Photonics Group, Department of Chemistry, University of Cambridge, Cambridge, UK. [2] Scripps Institution of Oceanography, University of California San Diego, San Diego, USA. [3] Marine Biological Section, Department of Biology, University of Copenhagen, Copenhagen, Denmark. [4] Department of Nanoengineering, University of California San Diego, San Diego, CA, USA. [5] Climate Change Cluster, University of Technology Sydney, Ultimo, Australia. [6] Department of Plant Sciences, University of Cambridge, Cambridge, UK. [7] These authors jointly supervised: Shaochen Chen, Silvia Vignolini. [8] Deceased: Mark Hildebrand. ✉email: dwangpraseurt@ucsd.edu; chen168@eng.ucsd.edu; sv319@cam.ac.uk

Powered by the high photosynthetic efficiency of the coral-algal symbiosis, coral reefs stand among the most productive ecosystems globally[1]. Photosynthetic performances in corals have been optimized by evolution in a competitive habitat with limited resources, leading to space efficient light management, high algal cell densities and photosynthetic quantum efficiencies that approach theoretical limits[2,3]. While different corals have developed a plethora of geometries to achieve such performances, they are all characterized by an animal tissue hosting microalgae built upon a calcium carbonate skeleton, that serves as mechanical support and as a scattering medium to optimize light delivery towards otherwise shaded algal-containing tissues[4,5]. Such algal self-shading is currently a key limiting factor for the upscaling of microalgal cultivation[6,7]. Therefore, fabricating bionic corals, where artificial biomaterials host living microalgae, can be pivotal for redesigning light management strategies for bioenergy and bioproduct generation[8,9].

Motivated by the optimized light management of corals, we have developed a bioprinting platform capable of 3D printing living photosynthetic matter mimicking coral tissue and skeleton source geometries. The hybrid living bionic corals are capable of cultivating high algal cell densities of up to $10^9$ cells mL$^{-1}$. Such findings open the way for coral-inspired biomaterials that can find use in algal biotechnology, coral reef conservation and in coral-algal symbiosis research.

## Results and discussion

**Development of 3D printed bionic corals.** Our bioprinting platform is capable of 3D printing optically-tunable photosynthetic material that mimics coral tissue and skeleton morphology with micron-scale precision (Fig. 1a–g). In principle, our technique allows replication of any coral architecture (Supplementary Fig. 1), providing a variety of design solutions for augmenting light propagation. Fast-growing corals of the family *Pocilloporidae* are particularly relevant for studying light management. Despite high algal cell densities in their tissues ($1 \times 10^6$ cells per cm$^2$ surface area), the internal fluence rate distribution is homogenous, avoiding self-shading of the symbiotic microalgae[10]. The photon distribution is mainly managed by the aragonite skeleton, where light leaks out of the skeleton and into coral tissue, supplying photons deep within the corallite[4,10]. In addition, light can enter the coral tissue more easily than it can escape, as low angle upwelling light is trapped by internal reflection due to refractive index mismatches between the coral tissue and the surrounding seawater[11]. We mimicked these light management strategies and designed a bionic coral made out of sustainable polymers for enhanced microalgal light absorption and growth.

To precisely control the scattering properties of the bio-inspired artificial tissue and skeleton, we developed a 2-step continuous light projection-based approach for multilayer 3D bioprinting (Methods). Optimization of the printing approach required a delicate balance between several parameters including printability (resolution and mechanical support), cell survival, and optical performance. The artificial coral tissue constructs are fabricated with a novel bio-ink solution, in which the symbiotic microalgae (*Symbiodinium sp.*) are mixed with a photopolymerizable gelatin-methacrylate (GelMA) hydrogel and cellulose-derived nanocrystals (CNC), the latter providing mechanical stability and allowed tuning of the tissue scattering properties[12,13]. Similarly, the artificial skeleton is 3D printed with a polyethylene glycol diacrylate-based polymer (PEGDA)[14] doped with CNC.

Based on optimization via experiments and optical simulations (Fig. 2a–j), the functional unit of the artificial skeleton is an abiotic cup structure, shaped like the inorganic calcium carbonate corallite (1 mm in diameter and depth) and tuned to redistribute photons via broadband diffuse light scattering (scattering mean free path = 3 mm between 450 and 650 nm) and a near isotropic angular distribution of scattered light (Fig. 2h, Supplementary Fig. 2), similar to the optical properties of the skeleton of fast growing intact corals[5,10]. The coral-inspired tissue has cylinder-like constructs (200 µm wide and 1 mm long) radially arranged along the periphery of the corallites mimicking coral tentacles, which serve to enhance surface area exposed to light[15] (Fig. 2a). We designed the bionic coral tissue to have a forward scattering cone (Fig. 2h), which enables light to reach the diffusely backscattering skeleton (Fig. 2a).

Our bionic coral increases the photon residence time as light travels through the algal culture (Fig. 2e). An increase in photon residence time (or photon pathlength) increases the probability of light absorption for a given density of microalgae[11]. As photons travel through the bionic coral, diffuse scattering by the bionic skeleton leads to an increasingly diffuse light field, effectively increasing the scalar irradiance (i.e., the integral of the radiance distribution from all directions around a point) as a function of tissue depth[4] (Fig. 2i, Supplementary Fig. 3). This photon augmentation strategy leads to a steady increase of scalar irradiance with tissue depth, which counterbalances the exponential light attenuation by photopigment absorption (Fig. 2g)[16].

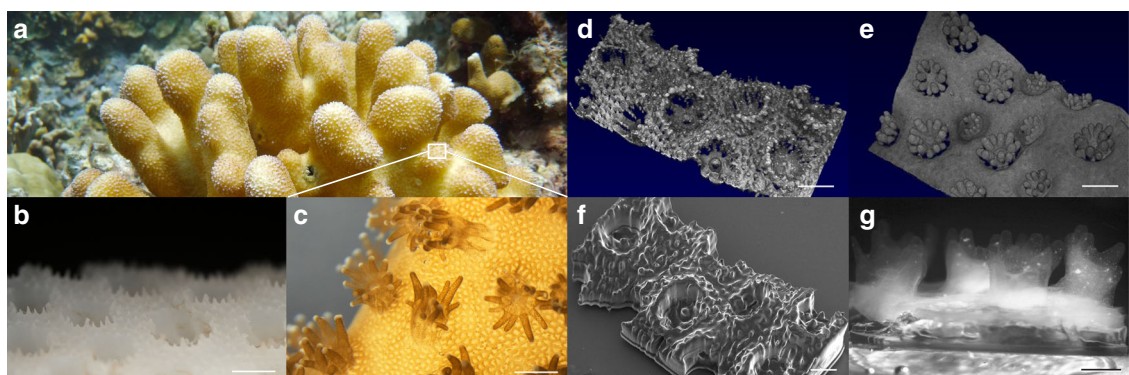

**Fig. 1 Structure of natural and 3D printed bionic corals.** Colony of the coral *Stylophora pistilla* growing at about 10 m depth on Watakobi Reef, East Sulawesi, Indonesia (**a**). Close-up photograph (**b, c**) and optical coherence tomography scanning (**d, e**) of coral skeleton and coral tissue, respectively. Scanning electron microscopy image of successful 3D printed skeleton replica showing corallites in 1:1 scale relative to the original design (**f**). Photograph of living bionic coral growing *Symbiodinium sp.* microalgae (**g**). The living tissue was printed on top of the skeleton mimic and the bionic coral was cultured for 7 days. Scale bar = 1 mm (**b–f**) and 750 µm (**g**).

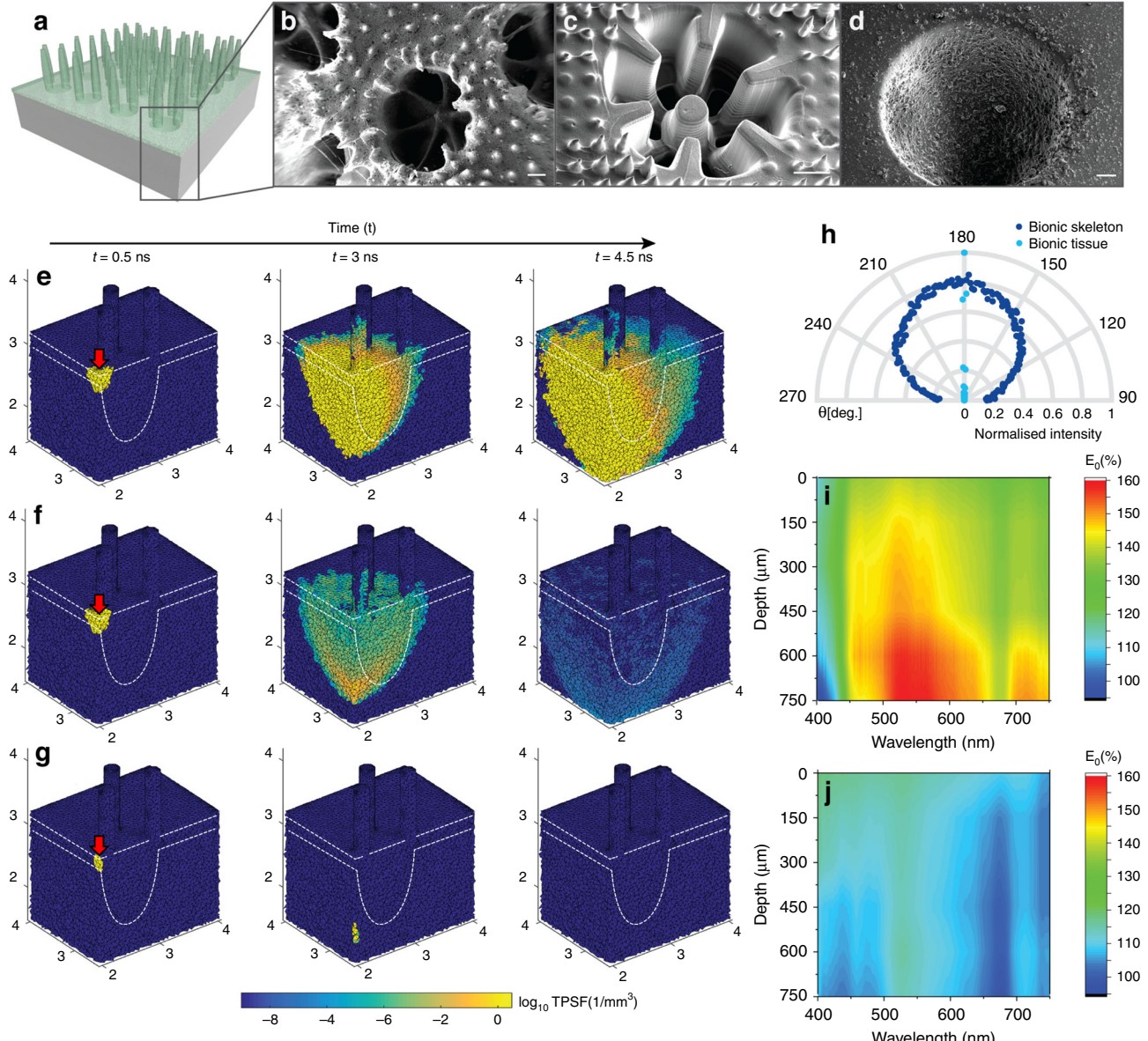

**Fig. 2 Optical properties of 3D printed bionic coral tissue and skeleton.** 3D rendering of final bionic coral design (**a**). Bionic skeletal design optimization (**b–d**) showing SEM images of the original *Stylophora pistillata* corallite architecture (scale bar = 200 μm) (**b**), a 3D printed intermediate skeleton design (scale bar = 300 μm) (**c**) and the final bionic skeleton doped with CNC aggregates (scale bar = 100 μm) (**d**). 3D tetrahedral mesh-based Monte Carlo simulation (**e–g**). Light (675 nm) is irradiated over the connecting tissue (red arrow) as a collimated pencil beam. The time-resolved solution of photon migration (temporal point spread function, TPSF [1/mm³]) is shown after 0.5 ns (left column), 3 ns (center column), and 4.5 ns (right column) in a cross-cut view of the 2-layer bionic coral (**e**), a 1-layer bionic tissue (**f**) and non-scattering GelMA (**g**). The microalgal density in the tissue component is identical for all simulations ($\mu_a = 15$ mm⁻¹). The angular distribution of forward scattered light ($\theta = 270°–90°$) at 550 nm is shown as normalized transmittance (**h**). Microprobe-based fluence rate measurements ($E_0$ in % of incident irradiance) for the bionic coral (**i**) and a flat slab of GelMA (**j**) both with a microalgal density of $5.0 \times 10^6$ cells mL⁻¹.

Compared to a flat slab of biopolymer (GelMA) with the same microalgal density ($5.0 \times 10^6$ cells mL⁻¹), the scalar irradiance (for 600 nm light) measured in the photosynthetic layer of the bionic coral is more than 1.5-fold enhanced at 750 μm depth due to the optimized scattering properties of the bionic coral tissue and skeleton (Fig. 2i, j, Methods). The bionic coral thus mimics the photon pathlength enhancement strategy that natural corals use to avoid algal self-shading[5,10,17].

**Performance evaluation of bionic coral.** In order to evaluate the growth of a free-living microalgal strain with a suitable fatty acid profile for bioenergy production we chose a green microalga of

the family Chlorellaceae (*Marinichlorella kaistiae KAS603*)[18] (Fig. 3a–d, Methods). We grew *M. kaistiae KAS603* under no-flow conditions and low incident irradiance ($E_d = 80$ μmol photons m⁻² s⁻¹) in our bionic coral, where it reaches algal cell densities of $>8 \times 10^8$ cells mL⁻¹ by day 12 (Fig. 3a). This is about one order of magnitude higher than the maximal cell densities reported for this algal species when grown in flasks under continuous stirring[18]. Despite such high algal cell densities, irradiance does not limit growth at depth, and about 80% of the incident irradiance remains at 1 mm depth within the bionic coral tissue construct (Fig. 3b). In comparison, standard biofilm-based photobioreactors are characterized by absorption dominated

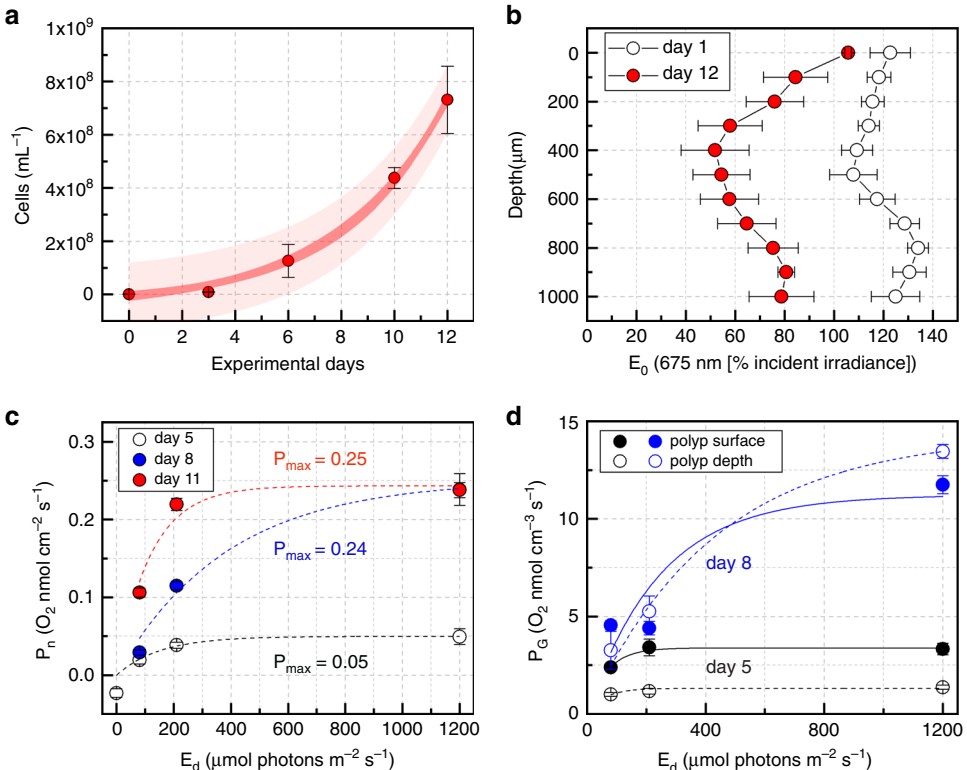

**Fig. 3 Performance testing of 3D printed bionic coral.** Growth of *Marinichlorella kaistiae KAS603* in bionic coral (**a**). Data are means ( ± SEM, $n = 3-6$ bionic coral prints). Dark and bright red areas show 95% confidence and prediction intervals, respectively. Vertical attenuation of fluence rate ($E_0$ at 675 nm) at the beginning (day 1, means ± SEM, $n = 4$) and end of the performance test (day 12, means ± SEM, $n = 3$) (**b**). Net photosynthetic rates at day 5, 8, and 11 (means ± SEM, $n = 3-10$ corallites) (**c**). Lines represent curve fits (see Methods). Gross photosynthetic rates at day 5 (black) and day 8 (blue) (**d**). Symbols are means (±SEM, $n = 2-6$ corallites), lines are curve fits. Measurements were performed with $O_2$ microsensors at the center of the corallite cup surface (closed symbols/solid lines) and at a vertical depth of 300 μm (open symbols/dashed lines).

exponential light attenuation leading to a virtual depletion of irradiance within 200–300 μm of the biofilm thickness[19]. We observed that *M. kaistiae KAS603* grows in the bionic tissue as dense aggregates (sphericity 0.75 ± 0.09 SD, diameter = 30–50 μm, $n = 44$ aggregates; Fig. 4a–d). Algal photosynthesis within the tissue construct yields a net photosynthetic $O_2$ production of 0.25 nmol $O_2$ cm$^{-2}$ s$^{-1}$ at the polyp tissue surface (Fig. 3c). Gross photosynthesis within 8-day old bionic coral polyps is enhanced at a depth of 300 μm compared to gross photosynthesis rates measured at the surface of the bionic coral tissue (Fig. 3d).

**Applications for algal biotechnology and coral research**. Our novel bio-ink shows excellent biocompatibility for both free-living and benthic algal strains (Figs. 3a, 4, Supplementary Fig. 4). In contrast to other biofilm-based systems that rely on natural algal settlement[20], the rapid 3D bioprinting process employed here allows for immediate encapsulation of different algal strains and supports their growth in a customized optical microhabitat. Currently, a direct comparison of our algal cultivation technique to commercial photobioreactors cannot be made, as we have limited the printing of our structures to the cm-scale, while commercial applications will depend on engineered, large scale, economically viable systems. However, 3D bioprinting is an additive manufacturing technique that is rapidly developing on industrial scales[21] and there is thus great potential for developing large scale bionic corals into spatially efficient photobioreactors for algal growth in e.g., dense urban areas or as life support systems for space travel[16,22].

Mechanistic studies on symbiont physiology and host microhabitat are fundamental for coral reef research but are currently hampered by the diversity and complexity of natural corals. We have shown that different coral host architectures can be successfully mimicked (Supplementary Fig. 1) which opens the way for controlled investigations on cellular activity of specific *Symbiodinium* strains, while mimicking the optical and mechanical microenvironment of different coral host species. Bionic corals also provide an important tool for advancing animal-algal symbiosis and coral bleaching research, as our 3D bioprinting approach can be exploited for manipulative studies on the photophysiology and stress response of different microalgal strains under in vivo-like animal host conditions. We therefore anticipate that bionic corals will trigger novel fundamental biological studies, inspire the development of synthetic photosymbiosis model systems and lead to disruptive technologies for efficient photon augmentation for algal biotechnology.

## Methods
**Optical coherence tomography imaging**. To create a digital mask of natural coral surfaces, a spectral-domain (SD) optical coherence tomography (OCT) system (Ganymede II, Thorlabs GmbH, Dachau, Germany) was used to image living corals (Supplementary Fig. 1). The OCT system was equipped with a superluminescent diode (centered at 930 nm) and an objective lens (effective focal length = 36 mm) (LSM03; Thorlabs GmbH, Dachau, Germany) yielding a $z$-resolution of 4.5 μm and a $x$–$y$ resolution of 8 μm in water. The imaged coral species (*Pavona cactus, Stylophora pistillata, Pocillopora damicornis, Favites flexuosa*) were maintained at the Centre Scientifique de Monaco, and corals were imaged under controlled flow and irradiance conditions. For OCT imaging of bare coral skeletons, the living tissue was air brushed off the skeleton. The skeleton was carefully cleaned before imaging the bare skeleton in water. OCT scanning was performed under controlled light conditions provided by a fibre-optic tungsten halogen lamp (KL-2500 LCD, Schott

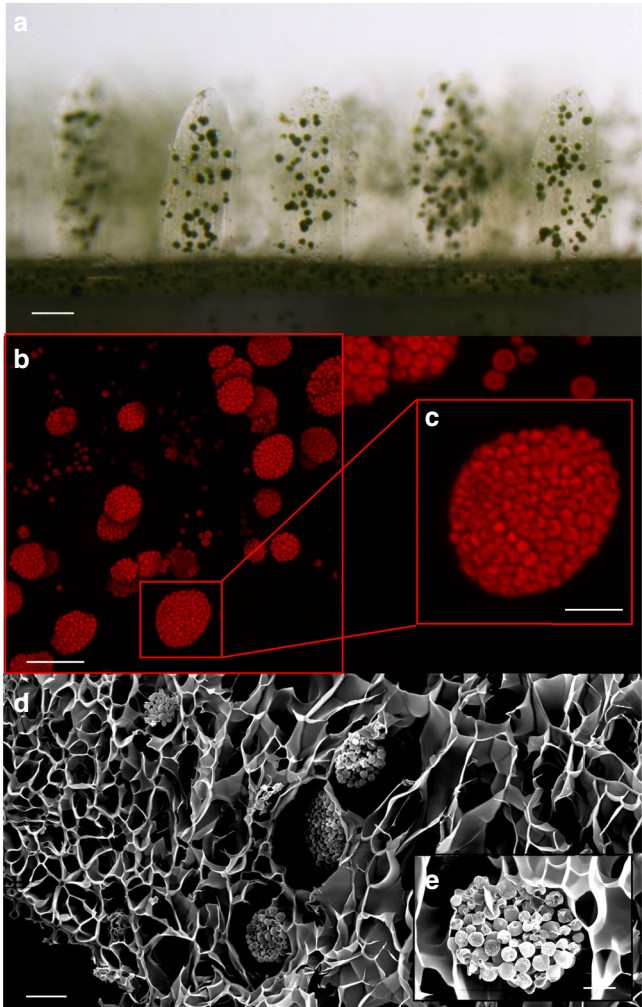

**Fig. 4 Living 3D printed bionic coral.** Horizontal view of 7-day old bioprinted construct, showing aggregates of the green microalga *Marinichlorella kaistiae KAS603* (scale bar = 100 μm) (**a**). Maximum *z* projection of confocal images showing chlorophyll *a* fluorescence of bionic tentacles (scale bar = 50 μm) (**b**) and a *M. kaistiae KAS603* aggregate (scale bar = 20 μm) (**c**). SEM image of bionic tissue showing porous tissue scaffolds (scale bar = 20 μm) (**d**) and a close-up of a microalgal aggregate (scale bar = 10 μm) (**e**).

GmbH, Germany) with coral samples placed in a custom-made acrylic flow chamber connected to a water reservoir providing aerated seawater[15].

**Surface rendering of OCT data.** OCT data was extracted as multiple 16-bit TIFF image stacks and imported into MATLAB (Matlab 2018a). Image acquisition noise was removed via 3D median filtering. Segmentation of the outer tissue or skeletal surface was done via multilevel image thresholding using Otsu's method on each image of every TIFF stack. The binary files were exported as *x,y,z* point clouds and converted to a *stl* file format, which could be sliced into 2D image sequences for bioprinting[23]. If the generated *stl* files showed holes in the surface mesh, these holes were manually filled using *Meshlab* (Meshlab 2016).

**Algal-biopolymer design optimization.** Key characteristics to achieve in material design were (1) high microalgal cell viability and growth, (2) microscale printing resolution, and (3) optimization of light scattering and biomechanical parameters including material stiffness, porosity and molecular diffusion. The photo-induced, free radical polymerization mechanism underlying our 3D printing technique allowed us to precisely control the mechanical properties via modulating the crosslinking density of the polymerized parts[24]. Any material and fabrication parameters (e.g., light intensity, exposure time, photoinitiator concentration, material composition) that affect the crosslinking density can be employed to tune the mechanical properties of the printed parts. Initially, different concentrations of prepolymer and photoinitiator combinations were evaluated, including glycidal

methacrylate-hyaluronic acid (GM-HA), gelatin methacrylate (GelMA), poly-ethylene glycol diacrylate (PEGDA), and poly(lactic acid), together with the photoinitiators Irgacure 651 and lithium phenyl-2,4,6 trimethylbenzoylphosphinate (LAP). We combined PEGDA with GelMA to make a mechanically robust and tunable hydrogel (Supplementary Fig. 5). GelMA was initially doped with graphene oxide, which enhanced mechanical stability but limited light penetration and cell growth. We developed a photopolymerization system using 405 nm light to avoid UV damage to the algae.

To optimize light scattering, we first mixed the hydrogel with different concentrations of $SiO_2$ particles (Sigma-Aldrich, USA) that were in a size range (about 10 μm) to induce broadband white light scattering with high scattering efficiency. However, when mixed into the hydrogels, the $SiO_2$ particle showed a vertical concentration gradient related to the particle sinking speed in the gel. Instead, we used cellulose nanocrystals (CNCs), which exhibit suitable light scattering, mechanical properties and low mass density[12]. CNCs can be considered as rod-shaped colloidal particles (typical length of 150 nm and a width of a few nm in diameter), which have a high refractive index (about 1.55 in the visible range). CNCs have received an increasing interest in photonics, due to their colloidal behavior and their ability to self-assemble into cholesteric optical films[25]. In the 3D bioprinted coral skeleton samples that contain 7% CNCs (w/v), we found that CNCs aggregated to form microparticles with a size range of 1–10 μm. These aggregated microparticles are highly efficient white light scatterers (Supplementary Fig. 2a). In contrast, the 3D bioprinted bionic coral tissue constructs contained only 0.1% CNCs (w/v), and we did not observe any aggregated CNC microparticles.

The printing polymer solution (bio-ink) for the bionic coral tissue constructs was made up of final concentrations of: *Marinichlorella kaistiae KAS603* ($1 × 10^6$ cells mL$^{-1}$), GelMA (5% w/v), LAP (0.2% v/v), food dye (1% v/v), PEGDA (6000 Da; 0.5% w/v), CNC (0.1% w/v), and artificial seawater (ASW; 93.2%). The yellow food dye (Wilton® Candy Colors) was added to limit the penetration of polymerization-inducing light into the bio-ink. This leads to higher light absorption relative to scattering and enhances the spatial resolution of the printing[24].

**Polymer synthesis.** PEGDA (mol wt, $M_n = 6000$) was purchased from Sigma–Aldrich (USA). GelMA was synthesized by mixing porcine gelatin (Sigma Aldrich, St. Louis, MO, USA) at 10% (w/v) into ASW medium (see above) and stirring at 60 °C until fully dissolved[25]. Methacrylic anhydride (MA; Sigma) was added until a concentration of 8% (v/v) of MA was achieved. The reaction continued for 3 h at 60 °C under constant stirring. The solution was then dialyzed against distilled water using 12–14 kDa cutoff dialysis tubing (Spectrum Laboratories, Rancho Dominguez, CA, USA) for 7 days at 40 °C to remove any unreacted methacrylic groups from the solution. The GelMA was lyophilized at −80 °C in a freeze dryer (Freezone, Labonco) for one week to remove the solvent[25]. CNC suspensions were prepared from the hydrolysis of Whatman cellulose filter paper (No.1) at 64 °C for 30 min with sulfuric acid (64 wt %), prior to quenching with ice water (Milli-Q)[12]. To obtain a stable suspension of CNC (2.2 wt %), the solution was centrifuged for 20 min at 20,000 × *g* and dialyzed against DI water (MWCO 12–14 kDa membrane). Acid and soluble cellulose residues were removed. The suspension was tip-sonicated in an ice bath (Fisher Ultrasonic) and vacuum-filtered using nitrocellulose filter (8.0 μm then 0.8 μm pore size, Sigma-Aldrich). The CNC suspension was evaporated under ambient temperature[24]. The photo-initiator lithium phenyl-2,4,6 trimethylbenzoylphosphinate (LAP) was synthesized in a two-step process[26]. First, 2,4,6-trimetyhlbenzoyl chloride (Sigma-Aldrich) was added slowly to an equimolar amount of dimethyl phenylphosphonite (0.018 mol, Acros Organics) via a Michaelis-Arbuzov reaction at room temperature and under argon. The mixture was continuously stirred for 18 h. Lithium bromide (6.1 g) in 100 ml of 2-butanone was carefully added and the solution was heated to 50 °C for 10 min. The resulting precipitate was filtered with 2-butonone under vacuum[26]. The product (LAP) was freeze dried and stored until further use. Freeze dried LAP was dissolved with ASW, and CNC was dispersed in the LAP solution via vortexing for about 5 min.

**Continuous multilayer 3D bioprinting of bionic coral.** The bionic coral design was developed as an optimization between algal growth rates, optical performance and the outcome of optical models (Fig. 2, Supplementary Figs. 2, 3). The final bionic coral was designed in CAD software (Autodesk 3ds Max, Autodesk, Inc, USA) and was then sliced into hundreds of digital patterns with a custom-written MATLAB program. The digital patterns were uploaded to the digital micromirror device (DMD) in sequential order and used to selectively expose the prepolymer solution for continuous printing. A 405-nm visible LED light panel was used for photopolymerization. A digital micromirror device (DMD) consisting of 4 million micromirrors modulated the digital mask projected onto the prepolymer solution for microscale photopolymerization[24]. The continuous movement of the DMD was synchronized with the projected digital mask to create smooth 3D constructs that are rapidly fabricated without interfacial artifacts. To print the bionic coral, a 2-step printing approach was developed. In the first step, the PEGDA bio-ink was used to print the coral inspired skeleton. The resulting hydrogel was attached to a glass slide surface, washed with DI water and then dried with an air gun. In the second step, the algal cell-containing bio-ink for tissue printing was then injected with a pipette into the skeletal cavities in order to fill the air gaps. The gap-filled skeletal

print was repositioned at the identical spot on the bioprinter, and the bionic coral tissue mask was loaded. The z-stage was moved such that the surface of the skeletal print touched the glass surface of the bioprinter.

**Algal stock culture maintenance.** Two microalgal species were chosen for inclusion in 3D bioprinted polymers: dinoflagellates belonging to the genus *Symbiodinium* and the green alga *Marinichlorella kaistiae*. Stock cultures of *Symbiodinium* strains A08 and A01 (obtained from Mary Coffroth, University of Buffalo) were cultured in F/2 medium in a 12 h/12 h light:dark cycle under an irradiance (400–700 nm) of 200 µmol photons $m^{-2}$ $s^{-1}$. Wild type *M. kaistiae* strain KAS603[18] were obtained from Kuehnle AgroSystems, Inc. (Hawaii) and were cultivated at 25 °C in artificial seawater (ASW) medium[27] under continuous light from cool white fluorescent lamps (80 µmol photons $m^{-2}$ $s^{-1}$). Stock cultures were harvested during exponential growth phase for use in bioprinting.

**Culturing of bionic coral.** Bionic corals harboring *Symbiodinium sp.* or *M. kaistiae* KAS603 were cultured under similar conditions as the respective algal stock cultures (see above). Prior to bioprinting, the bio-ink for printing bionic coral tissue constructs was inoculated with cell densities of $1 \times 10^6$ cells $mL^{-1}$ from exponentially growing cultures. We performed growth experiments with 35 bionic corals harboring *M. kaistiae* KAS603. The bionic corals were transferred to 6-well plates filled with 3 mL of ASW medium[27] containing broadband antibiotics (penicillin/ streptomycin, Gibco) at a concentration of 1:1000. All prints were illuminated with an incident downwelling irradiance (400-700 nm) of 80 µmol photons $m^{-2}$ $s^{-1}$ provided by LED light panels (AL-H36DS, Ray2, Finnex) emitting white light. The prints were grown without mixing at 25 °C. The ambient growth medium was replenished at day 5 and day 10. Degradation of GelMA-based tissue occurred after about 10–14 days when bacterial abundance was kept low via antibiotic treatment. Such degradation kinetics can be advantageous for more easy harvesting of the highly concentrated microalgae that are contained within the hard PEGDA-based skeleton.

**Optical characterization of bionic coral.** The angular distribution of transmitted light was measured using an optical goniometer[28]. The samples were illuminated using a Xenon lamp (Ocean Optics, HPX-2000) coupled into an optical fiber (Thorlabs FC-UV100-2-SR). The illumination angle was fixed at normal incidence and the angular distribution of intensity was acquired by rotating the detector arm with an angular resolution of 1°. To detect the signal, a 600 µm core optical glass fiber (Thorlabs FC-UV600-2-SR) connected to a spectrometer (Avantes HS2048) was used. To characterize the optical properties, the total transmitted light was measured for different sample thicknesses using an integrating sphere[28]. The samples were illuminated by a Xenon lamp (Ocean Optics, HPX-2000) coupled into an optical fiber (Thorlabs FC-UV100-2-SR), and the transmitted light was collected with an integrating sphere (Labsphere Inc.) connected to a spectrometer (Avantes HS2048). In the case of the skeleton-inspired samples, where the light is scattered multiple times before being transmitted, the light transport can be described by the so-called diffusion approximation[29]. In this regime, the analytical expression, which describes how the total transmission ($T$) scales with the thickness ($L$) for a slab geometry, is given as[30]:

$$T = \frac{1}{l_a} \frac{\sinh\left(\frac{z_e \times l_t}{l_a}\right) \sinh\left(\frac{z_e \times l_t}{l_a}\right)}{\sinh\left(\frac{L + z_e \times l_t}{l_a}\right)} \qquad (1)$$

where $l_a$, $l_t$ and $z_e$ are the absorption length, the transport mean free path and the extrapolation length, respectively[29]. Here, $z_e$ quantifies the effect of internal reflections at the interfaces of the sample in the estimation of $l_a$ and $l_t$[31]. We quantified $z_e$ by measuring the angular distribution of transmitted light, $P(\mu)$, which is related to $z_e$ by the following equation[32]:

$$P(\mu) = \mu \frac{z_e + \mu}{\frac{1}{2} z_e + \frac{1}{3}} \qquad (2)$$

where $\mu$ is the cosine of the transmission angle with respect to the incident ballistic beam. The theoretical fit is shown in Supplementary Fig. 2C and led to a value of $z_e = (1.32 \pm 0.12)$. Once the extrapolation length was estimated, the values of $l_a$ and $l_t$ could be calculated with Eq. (1) (Supplementary Fig. 2d, e). This was done with an iteration procedure to check the stability of the fit[31]. In the bionic coral tissue, the scattering strength of the material is too low and the diffusion approximation cannot be applied. In this regime, the extinction coefficient can be estimated using the Beer-Lambert law (Supplementary Fig. 2f).

The refractive index ($n$) of the bioprinted bionic coral tissue was determined with the optical goniometer to characterize the Brewster angle ($\theta_B$). A half circle of the material was printed with a diameter of 2 cm and a thickness of $z = 5$ mm. The Brewster angle was calculated according to Snell's law:

$$n = \frac{\sin(\theta_i)}{\sin(\theta_r)} = \frac{\sin(\theta_i)}{\sin(\theta_{90-i})} = \tan(\theta_i) \qquad (3)$$

and Brewster's law:

$$\theta_B = \arctan \frac{n_2}{n_1} \qquad (4)$$

where $\theta_i$ is the angle of incidence, and $\theta_r$ is the angle of refraction. $n_1$ and $n_2$ are the refractive indices of the medium and the surrounding medium, respectively. For the coral-inspired tissue $\theta_B$ ranged between 54.0° and 55.0° yielding a refractive index of $n = 1.37-1.40$.

**3D Monte Carlo time-of-flight photon propagation modeling.** Tetrahedral meshes were generated via Delaunay triangulation using the MATLAB based program *Iso2mesh* that calls *cgalmesh*[33]. Meshing was performed with different mesh properties varying maximal tetrahedral element volume and Delaunay sphere size in order to optimize simulation efficiency. Settings were optimized for a Delaunay sphere of 1 (10 µm) and a tetrahedral element volume of 5 (50 µm). Generated tetrahedral meshes were used as source architecture for a mesh-based 3D Monte-Carlo light transport simulation (*mmclab*)[34]. The model uses the generated tetrahedral mesh and calculates photon propagation based on the inherent optical parameters, the absorption coefficient $\mu_a$ [$mm^{-1}$], the scattering coefficient $\mu_s$ [$mm^{-1}$], the anisotropy of scattering $g$ [dimensionless] and the refractive index $n$ [dimensionless][35]. The optical parameters were extracted via integrating sphere measurements (see above) and were used to calculate time-of-flight photon propagation in the bionic coral. The probe illumination was a collimated point source with varying source positions.

**Mechanical properties of bionic tissue.** The Young's modulus of the bionic coral tissue was evaluated with a microscale mechanical strength tester (Microsquisher, CellScale). Each sample was preconditioned by compressing at 4 µm $s^{-1}$ to remove hysteresis caused by internal friction. The compression test was conducted at 10% strain with a 2 µm $s^{-1}$ strain rate. Cylindrical constructs were 3D printed using the same bio-ink as used to print bionic coral tissue. The Young's modulus was calculated from the linear region of the stress–strain curve[24]. Three samples were tested, and each sample was compressed three times.

**Cell harvesting.** Cell density was determined at the beginning of the experiment (day 0) and then at day 3, day 6, day 10, and day 12 of the growth experiments. To determine cell density, the construct was removed from the growth medium, and any remaining solution attached to the construct was removed with a Kimwipe. Each construct was transferred to a 1.5 mL microfuge tube and the hydrogel was dissolved via adding 600 µL trypsin solution (0.25% Trypsin/EDTA) under incubation at 37 °C for 40 min[23]. This procedure removed the microalgal cells from the matrix allowing for cell counting via a haemocytometer. The accuracy of this approach was verified by printing known cell densities (from liquid culture) and comparing it to the trypsin-based estimates yielding a deviation of < 3%. However, the matrix itself is biocompatible and non-toxic and does not need to be removed to harvest algal biomass.

**O₂ microsensor measurements.** Clark-type $O_2$ microsensors (tip size = 25 µm, response time < 0.2 s; OX-25 FAST, Unisense, Aarhus, Denmark) were used to characterize photosynthetic performance of the bionic corals. Net photosynthesis was measured via linear $O_2$ profiles measured with $O_2$ microsensors from the surface into the overlying diffusive boundary layer[2]. The sensors were operated via a motorized micromanipulator (Pyroscience, Germany). The diffusive $O_2$ flux was calculated via Fick's first law of diffusion for a water temperature = 25 °C and salinity = 30 using a molecular diffusion coefficient for $O_2$ = 2.255 × $10^{-5}$ $cm^2$ $s^{-1(2)}$. Gross photosynthesis was estimated via the light-dark shift method[36]. A flow chamber set-up provided slow laminar flow (flow rate = 0.5 cm $s^{-1}$) and a fiber-optic halogen lamp (Schott KL2500, Schott, Germany) provided white light at defined levels of incident irradiance (400–700 nm) (0, 110, 220, and 1200 µmol photons $m^{-2}$ $s^{-1}$)[2]. Photosynthesis-irradiance curves were fitted to an exponential function[37].

**Fiber-optic microsensors.** The fluence rate (=scalar irradiance), $E_0$, within the bionic coral was measured using fiber-optic scalar irradiance microsensors with a tip size of 60–80 µm and an isotropic angular response to incident light of ±5% (Zenzor, Denmark). The sensor was connected to a spectrometer (AvaSpec-UL2048XL-Evo, Avantes). Fluence rate measurements were performed through the tissue at a vertical step size of 100 µm using an automated microsensor profiler mounted to a heavy-duty stand and operated by dedicated software (Profix, Pyroscience)[2]. Depth profiles were measured from the planar tissue surface (i.e., areas distant from the tentacles) into the center of the bionic corallite. Fluence rate was normalized to the incident downwelling irradiance, $E_d$, measured with the scalar irradiance sensor placed over a black light well at identical distance and placement in the light field as the surface of bioprinted constructs.

**Scanning electron microscopy (SEM).** SEM images were taken with a Zeiss Sigma 500 scanning electron microscope. Samples were prepared in two different ways. To image the bionic coral skeleton made of PEGDA, samples were dried at room temperature and sputter coated with iridium (Emitech K575X Sputter Coater). To image the bionic coral tissue made of GelMA, samples were snap frozen with liquid

nitrogen, and were then lyophilized in a freeze dryer (Freezone, Labonco) for 3 days. The overall shape could not be maintained, but microscale structures (such as micropores of GelMA) were well preserved. The samples were sputter coated with iridium (Emitech K575X Sputter Coater) prior to imaging on the SEM.

**Confocal laser scanning microscopy (CLSM).** To characterize microalgal aggregate size and distribution in 3D, a confocal laser scan microscope was used (Nikon Eclipse TE-2000U). Bionic corals were placed on a cover glass and imaged from below with a 641 nm laser. Confocal stacks of chlorophyll $a$ fluorescence were acquired using a pinhole size of 1.2 μm, a vertical step size for z-stacking = 1 μm, and a $x$, $y$ resolution of 0.6 μm. Particle segmentation and visualization of the data was performed in ImageJ and the NIS confocal elements software (Nikon). Particle segmentation was performed via manual thresholding of 229–4095 gray scale values, with a cleaning factor of 6× this eliminates smaller particles that are not aggregates), hole filling and a smoothing factor of 2×. The segmented particles were analyzed for surface area, volume and particle density per volume.

**Reporting summary.** Further information on research design is available in the Nature Research Reporting Summary linked to this article.

## Data availability

All data are available in the main text, the source data file, the supplementary data files and the figshare repository https://doi.org/10.6084/m9.figshare.11911197.v1. 3D printing designs are available as stl files.

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

## Acknowledgements

We thank Martin Tresguerres, Jennifer Smith, Michael Allen, Qianqian Fang, Bryan Zhu, Debra Quick Jones, Haixu Shen, Jeffrey Alido, Shania Schull, and Tressa Smalley for help with experimental and analytical work and Adelheid Kuhnle for providing algal cultures. Image credit in Fig. 1a is given to Gianfranco Rossi. Funding: This study was funded by the European Union's Horizon 2020 research and innovation programme (702911-BioMIC-FUEL, D.W.), the European Research Council (ERC-2014-STG H2020 639088, S.V.), the BBSRC David Phillips Fellowship (BB/K014617/1, S.V.), the National Institutes of Health (R01EB021857; S.C.), the National Science Foundation (1907434; S.C.), the Carlsberg Foundation (D.W., M.K.), and the Villum Foundation (00023073; M.K.).

## Author contributions

Conceptualized the study: D.W., S.V., D.D.D., M.H., A.G.S., M.D., S.C.; developed 3D printing approach: D.W., S.Y., S.C.; developed optical model and characterized optical properties: D.W., G.J., S.V.; designed and performed cultivation experiments: D.W., O.G., A.S.; performed imaging: D.W., S.Y.; provided materials: S.C., F.A., M.K., S.V., D.D.D.; D.W. All authors critically assessed the results and wrote the manuscript. M.H. passed away.

## Competing interests

The authors declare no competing interests.
