## [Peer Review File · Nature Communications]

Reviewers' Comments:

Reviewer #1:

Remarks to the Author:

I appreciate the authors' responses and manuscript improvements here. The paper is deserving and ready for publication in Nature Communications. I request one last change in the abstract:

"and disruptive approaches for coral reef conservation"

change to:

"and coral bleaching research"

Reviewer #2:

Remarks to the Author:

I have assessed the revised manuscript, and feel positive about the publication of this MS in Nature Communications. The authors have toned down a number of questionable statements and added new information and clarifications into the main text and SI.

Reviewer #3:

Remarks to the Author:

I have reviewed the previous version submitted to Nature. I still read the manuscript with excitement, and feel that it is a great inspiration of coral's skeletal and colony structure to acquire light for photosynthesis in depth. Self-shading is a significant limiting factor in algal mass production and a coral polyp design enhances light scattering nature of the "skeleton" and dividing the bioreactor into numerous small chambers.

As before, I have a couple of technical comments for consideration.

1. I have some issue about the argument that the system alleviates algal self-shading. As Fig. 3b shows, indeed light intensity decreases with depth first and then increases, due to the enhanced scattering property of the bionic device; however, the depth increases only from 0 to 1000 micrometers, or 0-1 mm. If the device is scaled up and the depth increases beyond 1 mm, would the optical property remain? Would the scaling up require printing more or larger "polyps"?
2. The abstract mentions that this design will allow for replicating both structural and functional traits of the coral-algal symbiosis. However, natural corals rarely are light limited, because stony corals live in shallow water and each coral endodermal cell typically contains no more than 2 symbiont cells. Can the authors explain more clearly in the paper how the device can help with coral-algal symbiosis research?
3. Since the algal cells are embedded in gels, the device may not be friendly to flagellated algae that cannot grow well in solid media. Can this issue be resolved in the device?

Reviewers' comments:

Reviewer #1 (Remarks to the Author):

I appreciate the authors' responses and manuscript improvements here. The paper is deserving and ready for publication in Nature Communications. I request one last change in the abstract:

"and disruptive approaches for coral reef conservation" change to: "and coral bleaching research"

Reply: We thank the reviewer for accepting the manuscript for publication. We have amended the abstract: "and disruptive approaches for coral reef conservation" has been replaced with 'novel approaches for coral reef research' as our 3D bioprinting approach can also be used for fundamental studies.

Reviewer #2 (Remarks to the Author):

I have assessed the revised manuscript, and feel positive about the publication of this MS in Nature Communications. The authors have toned down a number of questionable statements and added new information and clarifications into the main text and SI.

Reply: We thank the reviewer for their positive evaluation about the publication of this MS in *Nature Communications*.

Reviewer #3 (Remarks to the Author):

I have reviewed the previous version submitted to Nature. I still read the manuscript with excitement, and feel that it is a great inspiration of coral's skeletal and colony structure to acquire light for photosynthesis in depth. Self-shading is a significant limiting factor in algal mass production and a coral polyp design enhances light scattering nature of the "skeleton" and dividing the bioreactor into numerous small chambers. As before, I have a couple of technical comments for consideration.1. I have some issue about the argument that the system alleviates algal self-shading. As Fig. 3b shows, indeed light intensity decreases with depth first and then increases, due to the enhanced scattering property of the bionic device; however, the depth increases only from 0 to 1000 micrometers, or 0-1 mm.

Reply: In a non-scattering medium, light attenuation occurs exponentially according to Lambert-Beer's law. Such rapid attenuation can be seen in e.g. biofilm-based photobioreactors, where light attenuates to less than 10% over a distance of 200-300 micrometers (see e.g. Li et al. *Biotechnol. Bioeng.* **113**, 1046-1055 (2016).) In our case we have shown that we enhance irradiance availability over a vertical distance of 1 mm (a typical height of a coral polyp) (see e.g. Fig.2) so we clearly enhance light availability compared to other purely light absorbing biofilm-based systems. In natural corals, such enhancement is responsible for the superior photosynthetic energy efficiency of coral tissues (e.g. Brodersen et al. *J. R. Soc. Interface* **11**, 20130997 (2014)).

If the device is scaled up and the depth increases beyond 1 mm, would the optical property remain?

The inherent optical properties i.e. κ_s - the scattering coefficient [cm^{-1}] and κ_a - the absorption coefficient [cm^{-1}] will remain identical but the apparent optical properties such as the fluence rate (E_0) [$\mu\text{mol photons m}^{-2} \text{s}^{-1}$] within the polyp will be affected by the enhanced optical pathlength.

Would the scaling up require printing more or larger "polyps"?

We envision that scaling the system occurs by an optimization of the 3D arrangement of the polyps. In natural corals (e.g. *Pocillopora damicornis*) polyps are arranged radially along the branches, while the branches themselves radiate outwards. This creates a very high 3D surface area relative to the planar surface area and we are currently working on realizing such scaling. Future studies will also deal with optimizing the size of the polyps for such scaling purposes. In our previous revisions to *Nature* we had included some of these concepts. However, the reviewers suggested that these should be removed. We agree with that and refrain from including further details about scaling of the system in the present version for *Nature Communications*.

2. The abstract mentions that this design will allow for replicating both structural and functional traits of the coral-algal symbiosis. However, natural corals rarely are light limited, because stony corals live in shallow water and each coral endodermal cell typically contains no more than 2 symbiont cells.

Reply: While a large part of stony corals lives in shallow water (0-10 m), corals are also very abundant in waters down to 30 m and in some areas are even dominant in the mesophotic zone (> 50 m of water depth, e.g. in the Red Sea and Florida Keys, (e.g., Vermeij, Bak, 2002. *Marine Ecology Progress Series*, 233, pp.105-116., Khang et al. *Current opinion in environmental sustainability* 7 (2014): 72-81.). Light is a key limiting resource for corals, and light limitation over such depth gradients has been studied over many years. Recently, it has been shown that strong light gradients can also occur within the tissues of corals (Wangpraseurt et al. 2012 *Frontiers in microbiology*, 3, p.316.) and across the heterogeneous coral architecture (Kaniewska et al. 2011, *Journal of phycology*, 47(4), pp.846-860) – even in corals on shallow reef flats (Wangpraseurt et al. 2014, *Limnology and Oceanography* 59: 917–926). Additionally, space competition with other corals (e.g. overgrowth and shading by larger corals) adds to the complexity of light management on coral reefs.

3. Can the authors explain more clearly in the paper how the device can help with coral-algal symbiosis research?

Reply: The evolutionary success of coral reefs and their existence in nutrient poor environments has largely been attributed to the highly efficient coral-algal photosymbiosis. However, mechanistic studies on symbiont physiology and host microhabitat are currently hampered by the lack of suitable model systems and experimental tools. Different coral species have developed different architectures (both on the polyp and colony scale) to optimize light harvesting. Additionally, the material properties (skeletal and tissue) vary strongly in order to optimize light scattering and absorption (see e.g. Marcelino et al. 2013, cited in original manuscript). This creates unique light microenvironments within coral tissues, despite identical regimes of incident irradiance. Such unique light microenvironments are at the core of explaining bleaching susceptibility, photosynthetic energy efficiency as well as the genotypic and phenotypic distribution of Symbiodiniaceae (e.g. Marcelino et al. 2013, *PLoS ONE* **8**, e61492, Brodersen et al. 2014, *J. R. Soc. Interface* **11**, 20130997). However, detailed mechanistic studies have so far been limited by the complexity of the coral holobiont as animal and algal physiology interact in a complex

environment and confound direct observations. With our 3D bioprinting platform we are now able to mimic the physico-chemical microhabitat of Symbiodiniaceae within corals, allowing for controlled manipulative studies under *in vivo*-like conditions. This opens up a new range of studies where e.g. the photophysiology and stress response of different clades of Symbiodiniaceae can be studied for different coral host mimics in relation to different ambient stressors.

We have extended this discussion to the manuscript:

1. 107: Mechanistic studies on symbiont physiology and host microhabitat are fundamental for coral reef research but are currently hampered by the diversity and complexity of natural corals. We have shown that different coral host architectures can be successfully mimicked (Extended data Fig. 1) which opens the way for controlled investigations on cellular activity of specific *Symbiodinium* strains, while mimicking the optical and mechanical microenvironment of different coral host species. Bionic corals also provide an important tool for advancing animal-algal symbiosis and coral bleaching research, as our 3D bioprinting approach can be exploited for manipulative studies on the photophysiology and stress response of different microalgal strains under *in vivo*-like animal host conditions. We therefore anticipate that bionic corals will trigger novel fundamental biological studies, inspire the development of synthetic photosymbiosis model systems and lead to disruptive technologies for efficient photon augmentation for algal biotechnology.

4. Since the algal cells are embedded in gels, the device may not be friendly to flagellated algae that cannot grow well in solid media. Can this issue be resolved in the device?

Hydrogels are over 80% water and the stiffness of the gels can be affected by the details of the photopolymerization approach employed. This allows us to create hydrogels with varying degrees of elastic moduli (Zhu et al. 2018 *Mater. Today* **9**, 951-959(2018)). It is thus possible to create highly porous soft tissues that in principle allow for movement of flagellated cells. We have responded to a similar comment in the last rebuttal, where we highlighted that we encapsulated not only benthic but also planktonic species and that biofilm formation is not a requirement for our cultivation approach. We also note that such detailed studies remain to be done in the future. We would like to thank the reviewer again for her/his comments on the manuscript and hope that it is now acceptable for publication in *Nature Communications*.

Reviewers' Comments:

Reviewer #3:

Remarks to the Author:

I commend the authors for doing a great job in further improving the manuscript and providing convincing responses to review comments. I recommend acceptance of the manuscript in its current form for publication in Nature Communications.